# Ideal Feedstock and Fermentation Process Improvements for the Production of Lignocellulolytic Enzymes

**Attia Iram [1], Deniz Cekmecelioglu [2] and Ali Demirci [1,*]**

[1] Department of Agricultural and Biological Engineering, Pennsylvania State University, University Park, PA 16802, USA; axi52@psu.edu
[2] Department of Food Engineering, Middle East Technical University, Ankara 06800, Turkey; denizc@metu.edu.tr
* Correspondence: demirci@psu.edu

**Abstract:** The usage of lignocellulosic biomass in energy production for biofuels and other value-added products can extensively decrease the carbon footprint of current and future energy sectors. However, the infrastructure in the processing of lignocellulosic biomass is not well-established as compared to the fossil fuel industry. One of the bottlenecks is the production of the lignocellulolytic enzymes. These enzymes are produced by different fungal and bacterial species for degradation of the lignocellulosic biomass into its reactive fibers, which can then be converted to biofuel. The selection of an ideal feedstock for the lignocellulolytic enzyme production is one of the most studied aspects of lignocellulolytic enzyme production. Similarly, the fermentation enhancement strategies for different fermentation variables and modes are also the focuses of researchers. The implementation of fermentation enhancement strategies such as optimization of culture parameters (pH, temperature, agitation, incubation time, etc.) and the media nutrient amendment can increase the lignocellulolytic enzyme production significantly. Therefore, this review paper summarized these strategies and feedstock characteristics required for hydrolytic enzyme production with a special focus on the characteristics of an ideal feedstock to be utilized for the production of such enzymes on industrial scales.

**Keywords:** lignocellulolytic enzymes; cellulase; hemicellulase; lignocellulosic biomass; pretreatment; lignin modifying enzymes; enzyme production

## 1. Introduction

Fossil fuels are still consumed at alarming rates, even though lignocellulosic biomass is the most abundant carbon resource, which can be utilized for the production of biofuels [1]. Besides, fossil fuels, which are detrimental to both the environment and human health, are continuously depleting. The industrial infrastructure leans towards the use of fossil fuels after the industrial revolution of the last 250 years. In recent years, it has been speculated that more than 14% of the world's energy consumption can be met by using lignocellulosic biomass as an alternative to fossil fuel energy [2]. Therefore, it is estimated that lignocellulosic biomass can provide more than 27% of the world's transportation fuel needs by 2035 [3].

Implementation of lignocellulosic biomass is numerous. Currently, the main usage of lignocellulosic biomass is in the agriculture, forestry, and industrial sectors where it is being used for energy crops, forestry by-products, and wood industry residues [2]. However, lignocellulosic biomass can also be used to produce many other chemical and physical products with the help of a better understanding of its internal structure. Lignocellulosic biomass is usually composed of cellulose (40–60%), hemicellulose (10–40%), and lignin (15–30%) [1]. The cellulose and hemicellulose components can provide sugars for the production of bioethanol, which is one of the most utilized sources of renewable energy. The lignin portion of the lignocellulosic biomass can be used for production of value-added products and heat or steam for electricity production. However, all these applications

are not currently being utilized widely because of the challenges in the breaking down of lignocellulosic biomass into its respective usable components such as sugars and lignin.

Among various challenges that are associated with the usage of lignocellulosic biomass, the most prominent one is the use of energy-intensive pretreatment processes to break down the complex biomass into its respective usable components. It is widely accepted that more than 40% of the processing cost of lignocellulosic biomass used as energy products comes from the pretreatment of the biomass [4]. Furthermore, the processing steps can also be energy-intensive, which would ultimately make most of the biomass utilization scenarios inefficient. The most common pretreatment strategies are employed to separate the lignin portion of the biomass so that sugars in the cellulose and hemicellulose components can be released to further process for the relative end-use. Cellulose is mostly present as semi-crystalline microfibers and is the main component of the lignocellulosic biomass. It comprises D-glucose molecules linked by β-1,4-glycosidic bonds with a degree of polymerization ranging from 800 to 10,000 [1], while the hemicellulose mainly contains xylose with arabinose, galactose, and glucose in smaller proportions with a degree of polymerization of 50–600. On the other hand, lignin contains coniferyl, sinapyl, and coumaryl alcohols connected by C-O or C-C bonds. Cellulose fibers are bundled together by hydrogen bonding and van der Waals forces. Lignin and hemicellulose essentially act as the resin between the empty spaces of cellulose. All three components are linked by both physical and chemical interactions (Figure 1).

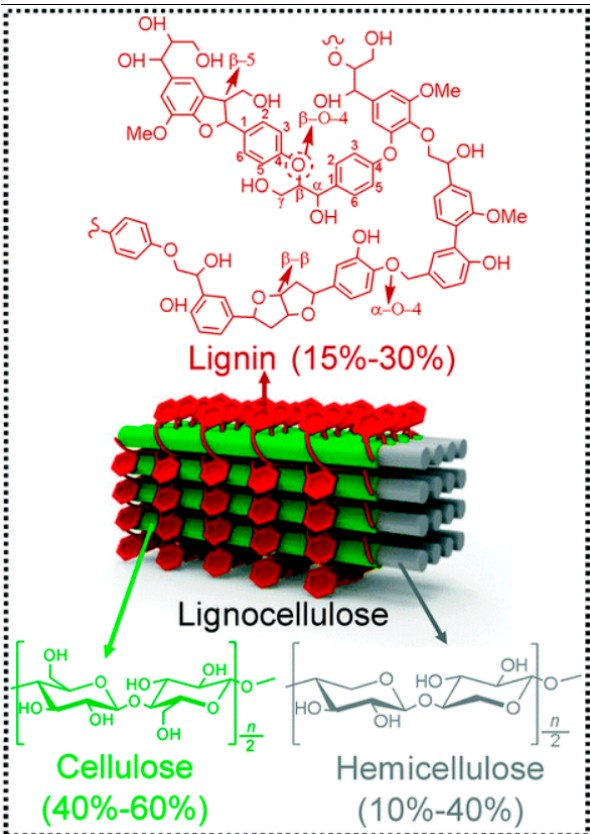

**Figure 1.** Three main components of lignocellulosic biomass, reproduced with permission from Wu et al., Photocatalytic transformations of lignocellulosic biomass into chemicals, published by Royal Society of Chemistry, 2020. [1].

In addition to the lignocellulosic components described above, there are many other molecular entities that can be present in a lignocellulosic biomass. Some examples of these molecular entities are proteins and their monomers (amino acids), pectin and pectin-like molecules, alkaloids, and inorganic molecules [5]. All these compounds can have different

effects on different types of microbial species. For example, a high molar ratio of carbon to nitrogen is needed for the microbial lipid production and accumulation [5]. On the other hand, an abundance of cellulose and hemicellulose fibers can also help in lignocellulolytic enzyme production [6].

Conversion of lignocellulosic biomass into biofuels such as ethanol and biogas can reduce the impact of greenhouse gases emitted from fossil fuels. However, the complex structure including the physical and chemical interactions between the cellulose, hemicellulose, and lignin, along with lignin acting as the physical barrier in the conversion, poses problems in establishing the industrial foundations in the use of lignocellulosic biomass. While the physicochemical methods such as heat, hot water, steam, etc. are effective in removing or excluding the lignin barrier, these energy-intensive methods cannot be controlled to liberate the sugar monomers in the cellulose and hemicellulose components [7]. Furthermore, the physical and chemical methods convert these sugars into chemicals that are then unusable or inhibit the microbial communities converting the biomass into biofuels. Therefore, the biochemical methods, i.e., hydrolytic enzymes, are employed to decrease the energy demands and production of unwanted inhibitory by-products during the conversion process.

Hydrolytic enzymes such as cellulases, hemicellulases, and lignases are produced by a wide variety of organisms including plants and microbial species such as fungi and bacteria. The rate of hydrolytic action along with overall efficiency in the degradation of the lignocellulosic biomass depends on many factors such as the source of the enzyme, substrate, and the number and type of the enzymes involved. For example, a set of different enzymes that must act in synergy is needed for the efficient hydrolysis of any given biomass. The external factors such as the cost of the enzyme production, quality of the enzyme cocktail, and type of pretreatment method required with the enzymatic hydrolysis also play an important role in the determination of the efficiency of the enzymatic conversion of lignocellulosic biomass into biofuels such as bioethanol and biogas.

Therefore, lignocellulosic hydrolytic enzymes are needed for efficient conversion of the lignocellulosic biomass into sugars that can be fermented into valuable products such as biofuels and other bioproducts. Since their discovery in the early 1950s, various enhancement strategies have been employed to increase the production, efficiency, and feasibility of the microbial hydrolytic enzymes [8]. The main barrier in making these enzymes industrially feasible is the cost and type of substrates involved in production of such enzymes. The research on optimization strategies for making these enzymes more competitive has been established for the last four decades. This review article summarizes those studies in terms of proposed feedstocks and fermentation enhancements for the production of the lignocellulosic hydrolytic enzymes. In addition, a special focus has been given to the characteristics of an ideal feedstock that can be utilized for production of these enzymes on an industrial scale. The challenges of obtaining such feedstock with the industrial scale of production have also been discussed.

## 2. Lignocellulosic Hydrolytic Enzymes

Different pretreatment methods have been proposed in the literature over the past three decades for different types of lignocellulosic biomass [1,9,10]. Among them, biochemical pretreatment, especially enzymatic breakdown, is not only environment-friendly, but also controlled on the molecular level by the type of enzymes used in the degradation, thus regulating the type of product obtained after the biochemical treatment. However, the current status of enzymatic hydrolysis is not well established because of the underlying economic issues in the industrial production of such enzymes. It is extremely important to understand the types and mode of action for such enzymes when they work alone or together on any type of lignocellulosic biomass.

Lignocellulosic hydrolytic enzymes can be roughly categorized based on the type of substrate they act on. Therefore, the most common three categories of these enzymes are cellulases, hemicellulases, and lignases, or lignin modifying enzymes. There can

be many subtypes of each category based on the organisms and culture conditions of their production [11]. Overall, it is critical to have an enzyme cocktail for the efficient degradation of lignocellulosic biomass on an industrial scale. However, if the aim is to have just one type of product, such as glucose from cellulose or xylose from the xylan fraction of hemicellulase, the enzyme proportions of the cocktails can be adjusted to obtain that product from the raw biomass. Table 1 gives a concise summary of the broader categories of lignocellulosic hydrolysis enzymes with their relative activity characteristics and areas of industrial applications. The following sections also summarize each of these lignocellulosic hydrolytic enzymes.

**Table 1.** Enzymes involved in the degradation of lignocellulosic biomass and their industrial applications.

| Enzyme | Type | Enzyme Commission (EC) Number | Activity-Characteristics | Application Areas | References |
|---|---|---|---|---|---|
| Cellulases | Endo-β-glucanase | 3.2.1.4 | Hydrolysis of 1–3 or 1–4 bonds in the beta-D-glucans within the chain | Cereal grains; polishing; feed supplements | [12,13] |
| | β-Glucosidase | 3.2.1.21 | Hydrolysis from non-reducing end | Flavor enhancement; biofuel industry | [12,14] |
| | Exoglucanases | 3.2.1.91 | reducing or non-reducing end creating cellobiose | Food, pulp and paper industry | [12,15] |
| Hemicellulases | Endo-β-1,4-xylanase | 3.2.1.8 | β-1,4 bonds within Xylan chains | Food industry | [16,17] |
| | 1,4-β-Xylosidase | 3.2.1.37 | Hydrolysis from non-reducing end in β-D-Xylan | Food industry | [17,18] |
| | Endo-1,4-β-mannosidase | 3.2.1.78 | β-1,4 bonds within mannan chains | Delignification in pulp industry | [17,19] |
| | 1,2-α-Mannosidase | 3.2.1.113 | Removal of terminal alpha-D-mannose residues | Delignification in pulp industry | [17,20] |
| | β-Mannosidase | 3.2.1.25 | Hydrolysis from nonreducing end to form d-mannose residues | Delignification in pulp industry | [17,21] |
| | α-Galactosidase | 3.2.1.22 | Hydrolysis of α-galactoglucomannan | Guar gum digestion | [17,22] |
| | β-Galactosidase | 3.2.1.23 | Hydrolysis of β-galactoglucomannan | Medicine; guar gum digestion | [17,22] |
| | α-L-Arabinofuranosidase | 3.2.1.55 | Hydrolysis of arabinoxylan and arabinoglucoronoxylan | Feed industry and baking | [17,22] |
| | α-Glucuronidase | 3.2.1.139 | Hydrolysis of arabinoglucoronoxylan | Food industry | [22,23] |
| | Acetyl esterase | 3.1.1.6 | Hydrolysis of the ester bond between arabinose and ferulic acid (Lignin) | Cider clarification | [17,22] |
| | Acetyl xylan esterase | 3.1.1.72 | Cleaving of Acetyl groups in hemicellulose | Cider clarification | [17,22] |

**Table 1.** *Cont.*

| Enzyme | Type | Enzyme Commission (EC) Number | Activity-Characteristics | Application Areas | References |
|---|---|---|---|---|---|
| Lignin modifying enzymes (LMEs) | Lignin peroxidase | 1.11.1.14 | Oxidoreductase | Waste treatment | [24] |
| | Manganese peroxidase | 1.11.1.13 | Oxidoreductase | Wastewater treatment in the production of synthetic dyes | [25] |
| | Phenoloxidases | 1.10.3.2 | Multicopper oxidases | Bioremediation | [26] |
| | Hybrid peroxidase | 1.11.1.16 | Oxidoreductase | Industrial waste treatment | [27] |

### 2.1. Cellulases

Cellulases are enzymes that can break down the cellulose portion of the lignocellulosic biomass to glucose. The significance of such enzymes in exploring the possible applications of lignocellulosic biomass started gaining research focus in the early 1950s [28]. The major focus has always been on the extraction of sugar monomers from cellulose, which can then be used for conversion into different value-added products such as ethanol. However, cellulose has a very compact and rigid structure that cannot be broken down into its simplest fermentable constituents with the help of just one enzyme. As shown in Table 1, there are three main constituents of cellulases simply because they can be activated by different regions of cellulose crystals [29]. These categories are exoglucanases, β-glucosidases, and endoglucanases.

Exoglucanases or cellobiohydrolases act at the end of the cellulose chains, producing mainly cellobioses. Some variants of the enzymes can also break down smaller chains of glucose, thus creating cello-oligosaccharides (Figure 2). The enzymes usually work by creating a substrate-binding tunnel and act effectively in the pH range of 4 to 5. The temperature range is from 37 to 60 °C, making them favorable to use with higher than normal incubation temperatures [30]. Different exoglucanases can act from both reducing and non-reducing ends of the cellulose chains, thus creating more synergistic opportunities for the efficient degradation of the cellulose chains. The two examples of such enzymes are CBHII/Cel6A (acts on the non-reducing end) and CBHI/Cel7A (acts on the reducing end) [30]. Both of these enzymes are produced by the same fungus (*Trichoderma reesei*), which makes it efficient in producing an enzyme cocktail with more efficient degradation capacity than other microorganisms. However, the main end-product is a disaccharide (cellobiose), which has an inhibitory effect on the cellulases. Therefore, strategies such as product removal from the enzymatic reaction mixture should be adopted while producing such enzymes.

Another strategy to decrease the inhibitory effect of cellobiose in the enzyme mixture is the addition of β-glucosidases. β-glucosidases are the cellulases that can degrade cellobiose and other smaller cellodextrins into glucose molecules. The glucose molecules can also have an inhibitory effect on cellulases in the mixture, but their inhibitory effect is less than that of cellobiose [30]. Some types of β-glucosidases can also have β-galactosidase activities, making it more efficient in the degradation of lignocellulosic biomass [30]. Therefore, β-glucosidases are categorized into three categories based on their activity location: (i) they can act intracellularly; (ii) by associating with the cell wall; or (iii) extracellularly [31]. The optimum temperature range for these enzymes is between 45 and 75 °C while their optimum pH depends on various factors including the activity location [30].

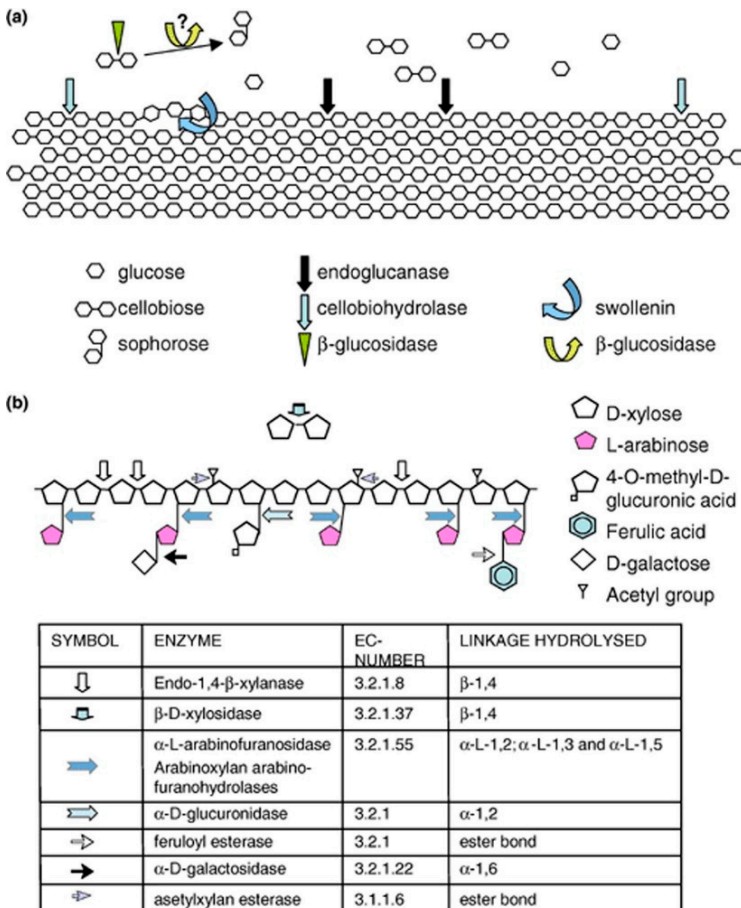

**Figure 2.** Schematic representation of cellulose and hemicellulose degradation by Aro et al., Transcriptional regulation of plant cell wall degradation byfilamentous fungi, published by Oxford University Press, 2020 [32]. (**a**): The degradation of cellulose; (**b**): the degradation of hemicellulose.

### 2.2. Hemicellulases

Unlike cellulose, hemicellulose is composed of heterogeneous polymer chains of variable lengths including pentoses (xylose, arabinose, etc.), hexoses (mannose, galactose, and glucose), and sugar acids. The variability in the polymer also gives a wide range of enzymes that are covered in the broader category of hemicellulases. Table 1 shows 11 different types of hemicellulases with their mode of action and current areas of application in the industry. Xylan is the most abundant constituent in the hemicellulose and comprises more than 70% of the different types of hemicellulose [30]. Therefore, endo and exo-xylanases are the most abundant hemicellulases in a given type of microbial substrate. The type of substrate can play an important role in composition of the hemicellulases [33]. For example, hardwood and grasses contain arabinoxylan and arabinoglucoronoxylan, respectively [22]. On the other hand, softwood contains galactoglucomannan in its hemicellulose. Based on these different substrates, different types of hemicellulases are produced, which will have varying pH and temperature ranges. However, if they are being produced by the same microorganism, they can act synergistically to give a wider degree of hydrolysis (Figure 2). Fungal species such as *Trichoderma* or *Aspergillus* species can produce more than 12 different types of hemicellulases by growing on different types of lignocellulosic carbon sources [34].

### 2.3. Lignases

Lignin, just like cellulose and hemicellulose, is a major component of lignocellulosic biomass. It is also the most abundant aromatic carbon source on earth. Lignin, with its complex aromatic structure, is more heterogeneous than cellulose and hemicellulose

components. The main purpose of lignin in vascular plants is to provide strength and rigidity by acting as a glue between different parts of cellulose and hemicellulose. The chemical structure of lignin (covalent interactions between phenylpropanoid units) makes it much more difficult to be degraded by the enzymes [35]. As shown in Table 1, the enzymes such as nonspecific oxidoreductases or manganese peroxidases more specifically play a role in the degradation of lignin [35]. These enzymes are secreted by different fungal species such as *Phanerochaete chrysosporium* [36]. Another enzyme, in this category, is lignin peroxidase (Figure 3). Lignin peroxidase receives more attention than manganese peroxidase in the literature as a degrader of lignin by fungal species [37].

**Figure 3.** Lignin degradation by lignin modifying enzymes, Timothy et al., The emerging role for bacteria in lignin degradation and bio-product formation, published by Elsevier, 2020 [38].

### 3. Applications of Lignocellulolytic Enzymes

The concept of using cellulases, hemicellulases, and lignin modifying enzymes to make value-added products from lignocellulosic biomass is relatively new as compared to the applications of such enzymes in other industries. Microbial hydrolytic enzymes have already been explored extensively for their applications in the food industry [39]. Table 1 represents some of the most common current applications of these enzymes in different industries. In case of cellulases, the food industry uses cellulases for wine, juice and bakery production, because cellulases provide improved wine filtration, improved maceration in the juice industry and improved texture and quality of bakery products, respectively [40]. Cellulases are also used in the paper, textile, and detergent industries [41]. For example, in the textile industry, the stone-washing of jeans is treated with pumice stone, which has undesirable effects on the final product, such as the decreased capacity of jean loading. The application of cellulases can produce 50% higher jean load [40]. In the juice industry, cellulases are a major component of macerating enzymes that help in the extraction, clarification, and stabilization of fruit juices. Cellulases are also used for the extraction of flavonoids from the seeds and flowers. Cellulases are effective in extraction as they ensure less heat damage and higher yields. Hemicellulases are used

in the bread-making industry as these enzymes are effective in the hydrolysis of the non-starch component of the flour. The rheological properties can be improved with the help of xylanases [41]. For example, a mixture of hemicellulases increases industrial degradation of various substrates by 12–109% [42]. On the other hand, hemicellulases can increase lignin removal from wood pulp by 27.8% [42]. Lignin modifying enzymes are usually used in industrial and commercial wastewater treatment (Table 1). Another application of such enzymes, especially phenoloxidases, is in bioremediation [26]. The majority of such enzymes are used in the form of enzyme cocktails with the combination of different enzymes for specific industrial applications. While all these hydrolytic enzymes have a wide range of applications in various industries, the major drawback in the use of such enzymes is their cost and handling conditions. They require specific pH and temperature ranges, which are not suitable for every application or sometimes require the design of additional reaction steps that can increase the cost of the process. However, such limitations can be decreased with the help of extensive research on the underlying enzymatic processes. This can ensure the higher efficiency and lower cost of the enzymatic reaction mixture and method.

### 4. Characteristics of an Ideal Feedstock

Industrial-scale production of any microbial product needs a feedstock that must have some definite economic, environmental, and process feasibility characteristics [43]. The microbial species should be able to utilize this feedstock to generate the required product or convert the feedstock into the respective product. There are many microbial species in nature that can grow on more than one type of feedstock [44]. However, different feedstocks have variable physicochemical natures, and their processing steps can also be different from each other. All such aspects play a huge role in the adaptability of a certain microbial process on an industrial scale.

One of the main characteristics of an ideal feedstock is the cost [45]. The overall economic impact of the microbial process can have many different types of costs such as capital cost, cost of the raw material, and the cost of pretreatment along with downstream processing. The cost of the raw material can be different for different types of feedstock. However, inexpensive feedstocks are always preferred over expensive ones. The category of feedstock can play an important role in this regard. For example, agricultural crops such as corn, wheat, and cotton are always more expensive than agricultural wastes such as sugarcane bagasse, rice straw, etc. [46]. Another factor in the selection of feedstock is the overall availability of feedstock [47]. Many inexpensive feedstocks might not be available for large scale processing due to their low rate of production. However, agricultural wastes are usually produced at higher rates and can be used for suitable microbial processes.

Another feature of an ideal feedstock is its production and location feasibility aspects. An ideal feedstock must be easy to produce and acquire. It should not need longer transportation times and should be stored until the appropriate time of usage. Most of the inexpensive and readily available feedstocks need some sort of preprocessing step. This step or procedure is usually known as pretreatment [48]. The pretreatment strategy should also be economical and environmentally friendly as it can drive the cost of the whole production process. The pretreatment process should also generate minimal amounts of hazardous by-products and gases.

While the cost, availability, and processing requirements are decisive characteristics of any feedstock, the microbial production of lignocellulolytic enzymes requires some additional features in their feedstocks. The microbial species that produce such enzymes can usually degrade complex carbohydrate sources into simple sugars and then use these sugars for growth and other metabolic processes. Such microbial species cannot degrade complex protein or lipid sources and therefore need simple sugars for their metabolic processes [49]. Therefore, the feedstock should contain different types of complex carbohydrates such as cellulose, hemicellulose, and lignin in some cases. The presence of lignin is usually considered problematic as it hinders access to cellulose and hemicellulose fibers,

thus decreasing the hydrolytic efficiency of the feedstock. Lignin, therefore, is usually removed with physical or chemical pretreatment strategies. Overall, typical lignocellulosic biomass is an ideal feedstock to produce lignocellulolytic enzymes.

## 5. Feedstocks for Lignocellulolytic Enzyme Production

Microbial strains that produce lignocellulolytic enzymes can utilize simple carbohydrates for growth [50]. However, if the culture media contain complex molecules such as starch, cellulose or hemicellulose, these microbial species induce the production of enzymes such as amylase, cellulase and hemicellulose, breaking the complex carbohydrates into their respective simple sugars, which can then be used for their growth [51]. Therefore, if the main goal is to produce carbohydrate degrading enzymes, it is imperative to add the respective carbohydrate in the media. There are many research articles that report the induction of a specific polysaccharide degrading enzyme by adding the respective polysaccharide in the media [52]. The same induction mechanism is used for the production of lignocellulolytic enzymes. These enzymes, specifically cellulase and hemicellulase, are produced by the microbial species capable of degrading such lignocellulosic feedstock in nature.

Among various lignocellulosic feedstocks that have been explored for the production of hydrolytic enzymes, corn stover, rice hulls, wheat bran, and sugarcane bagasse are the most common ones (Table 2). However, the production of such enzymes on different feedstocks is mostly different in research reports. For example, for corn stover, the enzyme production rate varies from 1.2 filter paper units per milliliter (FPU/mL) to 304 endoglucanase Units/g (Table 2). The most important difference here is the type of enzyme under study or the type of analysis used for the measurement of enzyme production rate and activity. Sugarcane bagasse is also one of the most commonly analyzed feedstocks in this regard. As can be seen in Table 2, the enzyme activities using sugarcane bagasse also depend on the type of analysis and the specific enzyme under study. However, these agricultural waste products produce an enzyme cocktail with a wide variety of different enzymes and all such enzymes can be detected in the culture media.

Enzymatic degradation of lignocellulosic biomass is gaining interest in both the research and industrial sectors because of its underlying applications in the energy and transportation industries [53]. Therefore, industrial-scale production of such enzymes from suitable microbial species has always been a topic of interest. The selection of microbial feedstock or the carbon source for such microbial species is one of the most prominent research questions since the discovery of lignocellulolytic hydrolytic enzymes [54].

**Table 2.** The feedstock analysis for the production of hydrolytic enzymes *.

| Year | Feedstock Used | Composition of Enzyme | Maximum Enzyme Produced | Units | References |
|------|----------------|------------------------|--------------------------|-------|------------|
| 1970–1979 | Feedlot waste | Cellulase and hemicellulase complexes | 0.4 enzyme cocktail | FPU/g/mL | [55] |
| | Ball-milled *Populus tremuloides* | Cellulase complexes | 1.5 enzyme cocktail | U/mL | [56] |
| | Wheat straw, sprouts, malt and corn cobs | Cellulase and hemicellulase complexes | 29.69 enzyme cocktail | mg sugar/mL media | [57] |
| | Sugar cane bagasse | Cellulases | 48.1 enzyme cocktail | % degradation of feedstock | [58] |

**Table 2.** *Cont.*

| Year | Feedstock Used | Composition of Enzyme | Maximum Enzyme Produced | Units | References |
|---|---|---|---|---|---|
| 1980–1989 | Tamarind kernel polysaccharide (TKP) | Cellulases, hemicellulases, β-glucosidase and β-xylosidase | N/A | N/A | [59] |
| | Kallar grass | CMCase and Xylanase | 3.8 CMCase 16.0 Xylanase | IU/mL | [60] |
| | Hemicellulose substrates and bagasse | Xylanase and xylosidase | 1.5 xylanase 0.08 β-xylosidase | U/mL | [61] |
| | Wheat bran, rice straw | β-d-glucosidase, d-xylanase | 60 B-d-glucosidase, 740 d-xylanase | U/g | [62] |
| | Kallar grass | CMCase, avicelase, xylanase, β-glucosidase and β-xylosidase | 3.2 CMCase | IU/mL | [63] |
| 1990–1999 | Corn cobs, rice hulls and melonseed shells | Cellulase and hemicellulase cocktails | 20 enzyme cocktail | % yield | [64] |
| | Corn Stover | Cellulase and hemicellulase cocktails | 0.7 FPA | nmol/mL/s | [65] |
| | Rice straw | Cellulases and xylanases | 9.7 FPA cellulase 9100 Xylanase | IU/l/h U/g | [66] |
| | Sweet sorghum silage | Cellulases and xylanases | 4 Cellulase 180 xylanase | IU/g | [67] |
| | Corn fiber | Cellulase and Xylanase | 3.4 cellulase 3.7 Xylanase | U/cm$^3$ | [68] |
| | Bagasse | Cellulase and β-glucosidase | 18.7 cellulase 38.6 β-glucosidase | IU/g | [69] |
| | Wood, straw | Cellulase | 7–18 | FPU/mL | [70] |
| | Barley and Wheat Straw | Xylanase, glucosidase, xylosidase, esterase and arabinofuranosidase | 0.16 cellobiase 1.4 xylanase | μmol/mL/min | [71] |
| | Orange peels | Cellulase, xylanase, pectinase | 3.39 cellulase 3.33 xylanase | U/mL | [72] |
| 2000–2009 | Corn stover | Cellulases and hemicellulases | 1.2 Filter paper activity | FPU/mL | [11] |
| | Rice straw and wheat bran | Cellulases and hemicellulases | 129 CMCase 100 β-glucosidase 5070 Xylanase | IU/g | [73] |
| | Sugar beet pulp | Endoglucanase, arabinosidase | 0.19 Endoglucanase 0.009 arabinosidase | U/mL | [74] |
| | Sugar cane bagasse | Cellulases and xylanases | 0.13 FPA 0.33 Xylanases | U/mL | [75] |
| | Wheat bran | Cellulolytic and hemicellulolytic enzymes | 1.05 endoglucanase 1.3 β-glucosidase 5.0 xylanase | U/mL | [76] |
| | Wheat bran and sugar cane bagasse | Cellulases and xylanases | 32·89 FPA 10 Xylanase | U/g | [77] |
| | Corn stover | Cellulolytic and xylanolytic enzymes | 304 endoglucanase 1840 Xylanases | U/g | [78] |

**Table 2.** *Cont.*

| Year | Feedstock Used | Composition of Enzyme | Maximum Enzyme Produced | Units | References |
|------|----------------|-----------------------|-------------------------|-------|------------|
| | Sorghum Bagasse | Cellulases and xylanases | 492.8 endoglucanase<br>297.8 Xylanases | U/g | [79] |
| | Wheat straw | Cellulases | 3.2 FPA83 CMCase | IU/mL | [80] |
| | Cassava waste | Cellulases | 0.46 CMCase<br>0.28 FPase | IU/mL | [81] |
| | Wheat bran and rice straw | Cellulases | 62.5 endoglucanase<br>3.0 FPase<br>196 Xylanase | units/g substrate | [82] |
| 2010–2020 | Horticulture waste | Cellulase and hemicellulase | 15 FPase<br>52.1 Xylanase | U/g | [83] |
| | Apple pomace | Cellulase and hemicellulase | 133.68 FPase<br>1412.58 Xylanase | IU/g | [84] |
| | Agricultural wastes | Cellulase and xylanase | 13.57 Cellulase<br>3106.34 Xylanase | IU/g | [85] |
| | Agricultural Wastes | Cellulase and xylanase | 30.22 FPase<br>427.0 Xylanase | U/g | [86] |
| | Apple pomace | Cellulase and hemicellulase | 383.7 FPase<br>4868 Xylanase | IU/g | [87] |
| | Sorghum and wheat bran | Cellulase and hemicellulase | 30.64 Cellulase<br>300.07 Xylanase | U/g | [88] |
| | DDGS | Cellulase and hemicellulase | 0.592 Cellulase<br>34.8 Xylanase | IU/mL | [89] |

* FPU: "Filter Paper Units", U: "Units", N/A: "Not Applicable", CMCase: "Carboxymethyl cellulase", IU: "International Units", FPA: "Filter Paper Activity", FPase: "Filter Paper cellulase", DDGS: "Distillers Dried Grains with Solubles".

By the principle of induction of enzyme production with the help of the respective form of polysaccharide in the media, it is ideal to include pure forms of such polysaccharides in the media. For example, cellulase production can be enhanced for most of the microbial species by adding pure cellulose or crystalline cellulose into the media. One example in this regard is the research report published by Ghose et al. [29]. The microbial substrate in this study was microcrystalline cellulose powder (MCPP), which is a pure form of cellulose [29]. Higher amounts of cellulase were obtained by this strategy. Another study on the same principle was conducted by Cai et al. [31]. They used Avicel, which is also a form of microcrystalline cellulose, for the production of cellulase [31]. For the production of xylanase, pure xylan (e.g., from Birchwood) is an ideal component in the culture media [90,91].

There are several issues with the approach of using pure polysaccharides as the feedstock. Most importantly, it limits the variety of enzymes that can be produced in the same incubation period with the microbial species that can produce several different hydrolytic enzymes. Production of cellulase is higher with cellulose in the media and production of hemicellulases is higher in a media where hemicellulose is the abundant polysaccharide [22]. One example is the study conducted by Novy et al. [22] in this regard. Figure 4 shows the effect of different feedstocks such as wood, sugarcane bagasse, and corn stover versus pure celluloses such as Avicel and Solka-floc. As can be seen, the enzyme production is higher with the use of pure cellulose as compared to agricultural waste products. This type of enzyme cocktail is more economical to use instead of just one enzyme for the degradation of lignocellulosic biomass. Another issue with the usage of a pure polysaccharide for the industrial-scale production of lignocellulolytic enzymes is the

higher cost of the pure polysaccharides as compared to the inexpensive feedstocks such as agricultural residues or industrial byproducts [92].

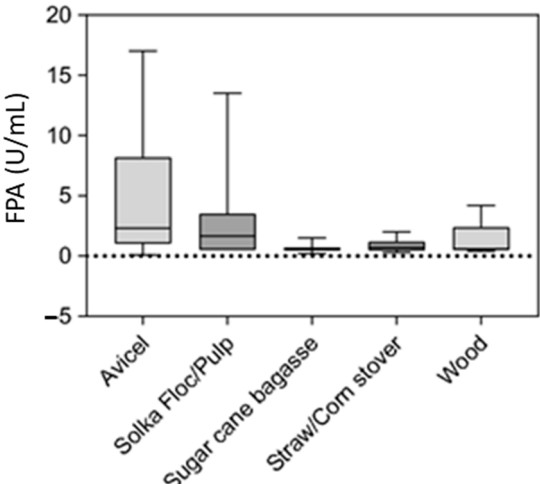

**Figure 4.** Production of cellulase (filter paper activity) by different types of microbial feedstocks, adapted from Novy et al. The influence of feedstock characteristics on enzyme production in Trichoderma reesei: a review on productivity, gene regulation and secretion profiles, published by Springer Nature [22].

Since the discovery of the concept of using lignocellulosic biomass as the energy source, there have been a large number of research reports depicting the potential microbial feedstocks for production of lignocellulolytic hydrolytic enzymes. As can be seen from Table 2, a large body of research has been undertaken over the past fifty years on the adaptability of different feedstocks for cellulase and hemicellulase production. Most of these research studies report a mixture of enzymes produced on the agriculture residues and waste products. However, the concentrations of the individual enzyme components have also been reported in most of the studies. With this body of research, it is easier to compare the effectiveness of different feedstocks in enzyme production. However, the main problem is in the enzyme analysis type and units of measurements. As can be seen from Table 2, most of the researchers adopt different approaches in the reporting of enzyme activities. Some examples of such measurements are FPU/g, IU/mL, U/g, or U/mL. While all such measurements depict the accurate results of enzyme production trends, these research results cannot be compared with each other without the conversion of measurement units. The factors such as moisture content in the feedstock and conditions of enzyme assays also play critical roles and results with different enzyme assay conditions cannot be compared.

Lignases or lignin modifying enzymes usually have different types of feedstock from cellulase and hemicellulase [93,94]. As has been mentioned in the earlier sections, the chemical structure of lignin is more heterogeneous than that of cellulose or hemicellulose. In addition, the physical and chemical interactions between phenylpropanoid units in lignin have a different mechanism of degradation. Therefore, the feedstocks for the production of lignin modifying enzymes can also be different from the general feedstocks that have been studied for cellulase and hemicellulase production [95,96]. Overall, the usage of agricultural residues is also explored for ligninolytic enzyme production [97,98]. The feedstock choice for the production of lignocellulolytic enzymes should favor the cheaper feedstock with the ability to produce enzyme cocktails that can degrade the lignocellulosic biomass effectively.

## 6. Pretreatment of Feedstock for Lignocellulolytic Enzyme Production

Pretreatment methods are developed to degrade the lignocellulosic structure in the biomass and make the cellulose and hemicellulose fibers available for microbial interaction.

However, different pretreatment strategies yield different types of outputs. Some of the desired outputs include low cost and higher amounts of reactive fibers. Overall, any given pretreatment method should have the following properties [99]:

- The pretreatment method should produce higher amounts of molecular entities for any specific product. For example, in the case of bioethanol production, the main ingredient required for microbial species is glucose or other monosaccharides which are to be converted into ethanol. Acid hydrolysis is one of the chemical methods that can produce higher amounts of simple sugars as compared to other methods [100]. On the other hand, if the final output is the production of lignocellulolytic enzymes such as cellulase, then the presence of reactive cellulosic fibers is required [101]. The physical pretreatment methods are usually employed to remove the lignin barrier so that cellulose and hemicellulose are available for subsequent biochemical reactions.
- The pretreatment method should not degrade the monosaccharides if they are the final product of the pretreatment. Some methods such as acid hydrolysis can degrade the pentoses and hexoses further into furfurals and Trihalomethanes (THMs), which may have an inhibitory effect on fungal activity [100].
- The pretreatment method should also not release any other type of compounds that can inhibit the growth of the microbial species, which are to be employed for the end-use of the lignocellulosic biomass. In this regard, all pretreatment methods should be checked and optimized according to the end-use of the products [102].
- The size and specifications of the pretreatment reactors should also be considered before employing a specific pretreatment method. For example, most of the acid hydrolysis reactions require high temperature and pressure (such as in an autoclave).
- The physical state of the pretreatment output also plays an important role in the determination of a suitable method. For example, if the liquid medium is required for the microbial or any other biochemical reaction, then the presence of solid residues at the end of the pretreatment procedure should be minimal.
- Simplicity of the pretreatment procedure is also required for setting up the reaction at different scales.
- The characteristics of the pretreatment feedstocks should also be taken into account before selecting a specific type of feedstock. For example, in the case of the sugarcane bagasse, chemical pretreatment methods along with steam or liquid hot water can be employed because it does not have high amounts of proteins or lipids. However, if the feedstock has high contents of proteins and lipids, such as distillers dried grains with solubles (DDGS), there is a high chance that such molecular components will also be degraded by the severe pretreatment conditions.

The ideal characteristics of a pretreatment method described above can vary in the specific product or the type of feedstock. There has been a tremendous amount of research dealing with the evaluation of different pretreatment methods for different types of products and feedstocks [48,100,103,104]. The most common trend is in the production of bioethanol and related biological products and biofuels [105,106]. The pretreatment methods for any type of feedstock for the production of lignocellulolytic enzymes have also been explored extensively (Table 3).

**Table 3.** Pretreatment types and examples for the production of lignocellulolytic enzymes *.

| Type | Pretreatment Method | Microorganisms | Example of Pretreatment Method | Feedstock | Enzyme Production | References |
|------|------|------|------|------|------|------|
| Physical | Liquid hot water | *Trichoderma reesei* | 200 °C for 30 min | Corn Cob | 3.5 FPU/mL | [107] |
| | Steam | *Trichoderma reesei* | 121 °C for 2 h | Horticultural Waste | 72 U/g | [83] |
| | Milling | *Trichoderma reesei* | Milled to 200 to 500 μm particle sizes | Horticultural Waste | 6.6 U/g | [83] |
| | Microwave | *Aspergillus heteromorphus* | 22.5 min irradiation time at 30 g/L substrate concentration | Rice Straw and Hulls | 14.1 U/g | [108] |
| Chemical | Dilute acid hydrolysis | 11 different bacterial and fungal strains | 5% sulfuric acid at 120 °C for 30 mins with 20% solid load | Distillers' Dried Grains with Solubles | 0.592 IU/mL | [89] |
| | Alkaline treatment | Endophytic *Acremonium* Species | 1% NaOH at 10% solid load | Sugarcane Bagasse | 0.14 U/mL | [101] |
| Biological | Fungal treatment | *Piptoporus betulinus* | 7 mm diameter mycelial discs | Rice Straw | 7.43 U/g | [109] |

* FPU: "Filter Paper Units", U: "Units", IU: "International Units".

As can be seen in Table 3, there are many different types of pretreatment methods employed for the production of lignocellulolytic enzymes. The physical pretreatment methods such as hot water and steam are employed at different temperature and time conditions. For example, Michelin et al. [107] employed the liquid hot water treatment method at different time periods and reported maximum production of hydrolytic enzymes after a 30-min treatment. On the other hand, the steam pretreatment method can give better results at shorter periods [89]. Sonication or the use of ultrasound waves for the treatment of biomass have also been explored recently to evaluate the potential in hydrolytic enzyme production. For example, a study conducted by Leite et al. [110] showed that the positive effect of ultrasound treatment was obtained for cellulase and xylanase production for solid state fermentation.

Table 3 also gives examples of chemical pretreatment methods such as acid or alkali treatment. Acid treatment usually comprises dilute sulfuric acid at higher temperature and pressure [100]. However, such methods should be optimized according to the production of byproducts that can hinder the microbial growth and production of desired products. Therefore, the extent of chemical pretreatment methods is usually determined not only by the degree of hydrolysis but the formation of undesirable products such as furfurals and the formation of desirable cellulose and hemicellulose fibers. Each method has its relative advantages and disadvantages and should be evaluated individually for enzyme production. Alkali methods are usually performed with the help of dilute bases such as ammonium hydroxide for prolonged time periods. The limitation of this method is the extensive time periods that are needed for the effective hydrolysis [100]. Other chemical methods such as organic solvent treatment are also being explored for the treatment of biomass [111].

On the other hand, there have been many studies where the two or more pretreatment methods are combined to give optimal results. One example, in this regard, is the grinding or milling of the feedstock before employing a physical or chemical method [83]. A combined chemical method is known as sequential acid and alkali treatment where the two most prominent chemical hydrolysis methods are combined in the biomass treatment. There are many studies that report the effectiveness of such methods for the production of simple sugars, which can then be used for the production of bioethanol or bio-butanol [112–114]. Such methods are promising for enzyme production as well since the release in the simple

sugars can complement the growth of the microbial species. Regardless of the type or conditions of the pretreatment method, the cost and environmental footprint should also be considered for each pretreatment method.

## 7. Microbial Production of Lignocellulolytic Enzymes

Among various sources of hydrolytic enzyme production, microbial fermentation has been studied most extensively because of the underlying economic and procedural applications [39,115]. The microorganisms involved in the production of such enzymes have been studied with a special focus on the number and types of enzymes produced with their optimum culture conditions. Microbial species involved in the degradation of lignocellulosic biomass can be found either free in nature or in the digestive tract of higher animals [34]. Many fungal and bacterial strains have gained special attention in terms of their industrial production of cellulases and hemicellulases [34,39,115]. Lignases, on the other hand, are mostly studied for their applications in the paint and synthetic dye industries and various microbial sources have been explored [116].

Fungal species are of special interest with regard to secreting cellulases and hemicellulases [30]. By the late 1970s, more than 14,000 fungal species have been identified and the number has been increasing ever since [30]. Among various genera of fungi, *Aspergillus*, *Trichoderma*, *Penicillium*, *Fusarium*, *Humicola*, and *Melanocarpus* species have been studied most extensively than others [117].

Among these, *Trichoderma* is of special interest due to various reasons. *Trichoderma* is one of the very few fungal strains that belongs to the category of "Generally Recognized as Safe" or GRAS [39]. *Trichoderma* species such as *Trichoderma reesei*, *Trichoderma atroviride*, *Trichoderma virens*, *Trichoderma lignorum,* and *Trichoderma harzianum* are best known for their enzyme production characteristics [39,118–120]. While various strains of these species are studied for their cellulolytic effects on various plants, *Trichoderma reesei* RUT-C30 has become a renowned name in the research on cellulase production. It has been more than thirty years since this strain has become a topic of various research articles for its cellulase production statistics [121]. *T. reesei* was first identified and named as *T. reesei* QM6a during world war II [122]. The hyperproduction of cellulases by this strain quickly gained interest after that and multiple research procedures were conducted to genetically modify this strain into a more efficient producer of cellulolytic enzymes. After a three-step mutagenesis procedure, QM6a was developed into RUT-C30 as one of the top cellulase-producing microorganisms [121].

*Aspergillus* is considered the second most prominent fungal genera for producing enzyme cocktails that can degrade lignocellulosic biomass effectively. Many *Aspergillus* species such as *Aspergillus niger* [73,88,123], *Aspergillus tubingensis* [124], *Aspergillus fumigatus* [125], and *Aspergillus oryzae* [126] have been recognized for cellulase and hemicellulase production in the last few decades. Genetic modification of *Aspergillus* species has also been conducted many times to make them more effective producers of lignocellulolytic enzymes [126,127]. While all these studies are examples of improved enzyme production for particular strains, problems such as the instability of mutant strains under different culture conditions and issues in the retention of mutation by the wild-type strains are common [121,128,129].

One of the most prominent challenges with using fungal strains for hydrolytic enzyme production is that these strains work best with solid-state fermentation (SSF) [130]. SSF has a problem in scaling up the production process and has not, therefore, been adapted at large scales so far [131]. Several bacterial species have been identified for lignocellulolytic enzyme production because they work best in submerged fermentation. Some examples of bacterial genera that are identified for cellulase production are *Bacillus*, *Clostridium*, *Pseudomonas*, and *Acidothermus* [117]. Among all these bacterial species, *Bacillus subtilis* is explored on a commercial scale for hydrolytic enzyme production [117]. On the other hand, thermophilic species such as *Thermomonospora fusca* are explored because hydrolytic

enzymes from these sources are used with high temperatures such as hot water and steam [132].

While looking for a microbial strain to produce lignocellulolytic enzymes, several factors must be taken into account. Some of these factors are production cost, scale, and period of culture development along with the possibility of genetic modification and other physical and chemical characteristics including the non-pathogenic nature of the species. All such factors can play a crucial role in the determination of a microbial strain that can be adopted for industrial-scale production. The desired characteristics of the product according to its uses should also be considered. For instance, one is the level of purity required for different uses of cellulase enzymes. Lower purity levels are acceptable for the biofuel industry while higher purity levels are required in the food industry. Such characteristics also play an important role in the determination of the type of strain to be used for enzyme production.

### 7.1. Modes of Fermentation for Lignocellulolytic Enzyme Production

Microbial production of hydrolytic enzymes, especially lignocellulolytic enzymes, is usually achieved with the help of fungal strains isolated from soils of decaying vegetation or compost of agricultural residues [133]. The fungal strains of these natural habitats usually prefer low moisture content in the substrate for growth and enzyme production. Such an environment is achieved in one fermentation mode known as solid-state fermentation (SSF). There have been many fungal lignocellulolytic enzymes produced on an industrial scale with the help of this fermentation mode. The SSF mode is applied in Japan to produce the enzymes from *T. koningii*, *T. viride*, and *T. reesei* with some *A. niger* strains as well [134]. Although there are claims that SSF is economically feasible and can be employed on larger scales for enzyme production, many issues with this mode of fermentation still exist, and the solution to these problems is provided by submerged fermentation (SmF), which is controlled more efficiently and does not have the scalability issues faced by SSF [43]. The submerged fermentation can also be employed for the enzyme production from bacterial species. All such factors make these two fermentation modes competitive to each other regarding relative advantages and challenges associated with each mode.

SSF is defined as fermentation without the availability of free water. However, the substrate for SSF should have enough moisture content to support microbial growth. Many fungal strains have been adapted for lignocellulolytic enzyme production using SSF at an industrial scale. One example, in this regard, is the domestication of *A. oryzae* for hydrolytic enzyme production [135]. SSF is better known for the production of mycelial patterns in such a way that both aerial and substrate hyphae are produced [136]. Solid-state fermentation ensures effective colonization of fungal mycelia within the solid substrate. To penetrate the solid substrate, fungal species often have to produce the enzymes that are necessary for the localized degradation of the substrate. As a result, higher yields of lignocellulolytic enzymes are achieved with the help of SSF.

Based on the type and design of the substrate, there can be two types of SSF to be employed on an industrial scale [137]. The solid-state substrate can either be an organic material that can be broken down by the fungal species with the help of enzymes or it can be made of inert material and a liquid layer of media spread over the inert material for microbial growth. The ideal substrate should not be dissolved in water, but it should absorb some water to comprise moisture for the microbial species.

There are several limiting factors that influence the development of optimum cultures of microbial species on SSF. Among various disadvantages presented by the SSF, the most prominent is the availability of water as a limiting factor. There have been many studies where problems with the standardization of enzyme production and growth have been reported with SSF as the production mode [43]. Another issue is the reproducibility of the acquired results. Because of the uncontrollable culture conditions in SSF, it is often not in the hands of researchers to predict the exact optimum culture conditions in an SSF environment. Another parameter that cannot be controlled fully in an SSF reactor is the

temperature. The rise of the temperature can then denature the produced enzymes, hence undermining the actual potential of the microbial strains for enzyme production [43].

To solve the scalability and the control issues, the submerged fermentation is often adopted by researchers on an industrial scale. In submerged fermentation, culture parameters are controlled more efficiently along with the type of carbon and nitrogen sources. The free available water is often considered the most advantageous aspect of the submerged fermentation as it helps in the efficient distribution of nutrients and temperature [133]. Many studies report the relative advantages of SmF for hydrolytic enzyme production over the conventional SSF [43]. The bioreactor designs with SmF are employed for the efficient control of pH and temperature along with agitation and aeration rates. There are many studies that show the effect of such culture parameters on enzyme production [138]. The feasibility of using bacterial species other than fungal for lignocellulolytic enzyme production has also been reported [138]. All such advantages make SmF a more attractive research topic as compared to SSF. However, SmF has its relative disadvantages and the most prominent one is the inability of longer incubation periods for fungal strains. The mycelial growth in the liquid is usually hindered by the limited size of the bioreactor, which, in return, will require more capital investment for SmF as compared to SSF [139]. There have been reports showing the requirement of 78% more capital investment in SmF than that of SSF [43]. The need for sophisticated equipment is another problem faced in SmF design. Therefore, it can be concluded that more research is needed to explore the potential of both fermentation modes for making the lignocellulolytic enzyme production process more economically feasible.

*7.2. Fermentation Enhancement Strategies*

Production of lignocellulolytic enzymes is one of the most prominent ways to degrade the lignocellulosic biomass into value-added products in the energy sector. However, the microbial production of such enzymes has experienced many limitations because of the underlying biochemical and process design issues. Over the last few decades, many research efforts have been developed to enhance microbial enzyme production for the sole purpose of the hydrolysis of the lignocellulosic biomass [60,133,140,141]. Most of these efforts involved the genetic modification of the microbial species or incorporation of the enzyme-producing gene in the organism so that it can be controlled more properly [142–145]. However, all such efforts have limitations in terms of industrial adaptations of genetically modified strains and the stability of the induced mutations [146].

Another effective approach to increase the production or activity rate of any enzyme is through the optimization of fermentation operation parameters. There are many studies that report the effect of culture parameters such as pH, agitation, temperature, and aeration rate on enzyme production [147,148]. The effect of these parameters cannot only change the overall protein production in the microbial media but also the enzyme activity levels per unit of volume or weight of the protein. Therefore, the optimized culture parameters can enhance enzyme production and decrease the overall cost of the production process, making them more applicable at an industrial scale.

Table 4 summarizes some of the studies conducted for the optimization of fermentation parameters for higher production of lignocellulolytic enzymes. The optimum pH range for enzyme production is mostly found between 4 and 7. The optimum temperature can go from 28 to 50 °C for mesophilic microbial species. However, there are studies that show the potential of thermophilic microorganisms with the ability to produce different types of cellulases and hemicellulases [149–151]. The relative advantage of using thermophilic enzymes is that they can be coupled with high-temperature pretreatment strategies without cooling down the reactor for the enzymatic reaction. However, the maintenance of such thermophilic species for industrial enzyme production is a relatively difficult process and is, therefore, not explored extensively. The increase in enzyme production with the help of culture optimization techniques can be seen in Table 4.

**Table 4.** Culture optimization techniques for lignocellulolytic enzyme production *.

| - | Optimized Conditions | | | | | | - |
|---|---|---|---|---|---|---|---|
| Enzyme | pH | Temperature (°C) | Agitation (RPM) | Time (h) | Microorganism (s) | Increase in Production | References |
| Cellulase | 4–4.5 | 28 | 120 | 96 | *Aspergillus niger* | 0.02 to 0.1813 IU/mL | [152] |
| CMCase | 7.2 | 39.11 | 121 | NO | *Bacillus subtilis* | 0.43 to 0.56 U/mL | [138] |
| Cellulase | 7.5 | 40 | NO | 96 | Different *Pseudomonas* and *Bacillus* species | 0.98–3.4 U/mL | [153] |
| Cellulase | 4 | 35 | NO | 54 | *Aspergillus niger* | 0–0.37 IU/mL | [147] |
| FPase | NO | 32.8 | NO | 144 | *Trichoderma viride* | 0.12 to 0.55 U/mL | [154] |
| Xylanase | NO | 34.7 | NO | 158 | *Trichoderma viride* | 30 to 145 U/mL | [154] |
| FPase | NO | 37 | NO | NO | *Aspergillus fumigatus* | 0–9.73 U/g | [155] |
| CMCase | 5.5 | 30 | NO | 264 | *Fomitopsis* sp. | 0–71.7 IU/g | [156] |
| CMCase | 5.5 | 50 | NO | NO | *Paenibacillus terrae* | 0.1–2.08 U/mL | [157] |
| Xylanase | NO | NO | NO | 264 | *Schizophyllum commune* | 0.08–5,740 IU/mL | [158] |

* IU: "International Units", U: "Units", NO: "Not Optimized".

Agitation rate and incubation time are two other factors that have been optimized extensively for the maximum production of hydrolytic enzymes [138,152,153]. While agitation is required for the effective mixing of nutrient and uniform temperature as well as providing higher oxygen transfer rates, it can also disrupt the mycelial growth in fungal cultures. The higher rates of agitation also require higher energy input. All such factors should be considered for the industrial production of the enzymes. The incubation time determines the highest activities during the growth cycle of the microbial species. The optimization of all such culture parameters can ensure the effective production of enzymes, which can be both economical and more environmentally friendly due to less energy inputs.

In addition to the culture parameters, the nutrient components can also greatly affect the specific enzyme production. The impact of feedstock has already been discussed in Sections 4 and 5 of this review. The choice of feedstock can be considered one of the most prominent factors in the determination of the effectiveness of any microbial production process. The feedstock does not only provide support and a carbon source for the SSF, but it can also provide other essential nutrients as well. One example in this regard is the use of DDGS as the feedstock for enzyme production. DDGS is a coproduct of industrial bioethanol production and has a high amount of cellulose and hemicellulose fibers, proteins and amino acids that can act as a nitrogen source for the microbial growth and production of enzymes [89]. There are, however, many other lignocellulosic feedstocks such as sugarcane bagasse or wheat straw that are lower in nitrogen content. In such cases, the addition of an inexpensive nitrogen source can enhance enzyme production. Table 5 shows some examples of research setups, where the optimization of such nitrogen sources has been conducted for maximum enzyme production. The variety in the studied nitrogen sources is worth mentioning in this regard. Relatively more expensive nitrogen sources such as yeast extract along with cheaper sources such as urea are tested for their impact on hydrolytic enzyme production (Table 5).

**Table 5.** Nitrogen source optimization for lignocellulolytic enzyme production *.

| Enzyme | Nitrogen Source Optimization | Microorganism (s) | Increase in Enzyme Activity | References |
|---|---|---|---|---|
| Cellulase | 0.125% peptone | *Aspergillus niger* | 0.05–0.1813 IU/mL | [152] |
| Cellulase | 4 g/L NaNO$_3$ | *Penicillium occitanis* | 0.5–13 U/mL | [159] |
| FPase | 3% sulfite pulp | *Trichoderma viride* | 0.12–0.39 U/mL | [154] |
| Xylanase | 3% sulfite pulp | *Trichoderma viride* | 30–70.25 U/mL | [154] |
| FPase | 0.25% beef extract | *Aspergillus fumigatus* | 0–9.73 U/g | [155] |
| FPase | 80.2 g/L Peptone | *Sclerotium rolfsii* | 0–5.72 FPU/mL | [141] |
| Xylanase | 55.4 g/L yeast extract | *Schizophyllum commune* | 0.08–5.74 IU/mL | [158] |

* IU: "International Units", U: "Units", FPU: "Filter Paper Units".

The optimization of SSF is more difficult to control and standardize as compared to SmF. The critical parameters in SSF are moisture content of the solid substrate, particle size of the substrate, pH, temperature, and aeration [160]. The control of such parameters is designed according to different reactor designs [160]. Overall, the optimization of culture parameters and nutrient sources can help extensively in achieving higher production of lignocellulolytic enzymes. Such strategies cannot only decrease the cost of the production processes but can also enhance the adaptability of the use of lignocellulosic biomass as the energy source to replace fossil fuels.

### 8. Challenges of Enzyme Production

For efficient utilization of one of the largest carbon sources on earth, such as lignocellulosic biomass, hydrolytic enzymes are needed to degrade such material. However, the high cost and technological limitations are still barriers to the commercialization of such lignocellulosic biomass utilization. The main requirement is the development of production processes, which are not only economical, but also adaptable on larger scales. In the current body of research, there are numerous studies reporting the enhancement strategies, which are based on culture optimization and genetic modification. However, the adaptation of such strategies at an industrial scale is always challenging [161]. The scaling up of the enzyme production technologies is difficult. One example is the enzyme activity rates that are influenced by the size of the reactor [162]. The parameters such as oxygen mass transfer and agitation rate are not easily controlled while scaling up the production process. Such problems are more prominent in the case of aerobic microbial species, which are also good producers of lignocellulolytic enzymes.

Another issue is the variable characteristics of feedstocks for different microbial processes and species. The same microbial species can show different production rates due to the variability in the feedstock characteristics. One example of such characteristics is moisture content [160]. Another issue is the availability of ideal feedstock on the site of enzyme production. Often, the site of the production of feedstock can be very far from the enzyme production facilities. In such cases, the life cycle analysis can determine whether the enzyme production and the subsequent application procedure of such enzymes are economically feasible or not. The overall carbon footprint of such enzymes is also another factor that must be minimized before the implementation of enzyme production. After five decades of research on the production process of enzymes, such challenges are still present, and more research is underway to minimize the impact of these issues.

### 9. Conclusions and Future Perspective

Hydrolytic enzymes such as cellulases, hemicellulases, and lignases are the key to the effective utilization of lignocellulosic biomass on earth. More than five decades ago, the role of such enzymes in the degradation of the world's most abundant carbon source was discovered. Since this discovery, research on the production of such enzymes has been

conducted. The effect of feedstock and its pretreatment strategies has also been studied extensively. The optimization of culture conditions and the nutrient elements in the media are also the focus of current research setups. However, even after the development of a large body of research dealing with all these aspects, limitations in the adoption of enzyme production at large industrial scales are still present. These limitations can be dealt with by acquiring more knowledge about the enzyme production process and the feedstocks available for this process. Therefore, future research should focus on the development of production methods that utilize inexpensive and readily available feedstocks, which also require minimal pretreatment energy.

**Author Contributions:** Conceptualization, A.D. and D.C.; validation, A.I., A.D. and D.C.; formal analysis, A.D. and D.C.; investigation, A.I.; resources, A.D.; writing—original draft preparation, A.I.; writing—review and editing, A.D. and D.C.; supervision, A.D.; project administration, A.D.; funding acquisition, A.D. All authors have read and agreed to the published version of the manuscript.

**Funding:** This research was funded by United States Department of Agriculture (USDA) National Institute of Food and Agriculture Federal Appropriations, Project PEN04594 and Accession number 1007291.

**Institutional Review Board Statement:** Not applicable

**Informed Consent Statement:** Not applicable

**Data Availability Statement:** The data presented in this study are available on request from the corresponding author.

**Acknowledgments:** This work was supported in part by the FULBRIGHT Student Program by providing a scholarship to Attia Iram.

**Conflicts of Interest:** The authors declare no conflict of interest.

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
