# Peer review of "Ideal Feedstock and Fermentation Process Improvements for the Production of Lignocellulolytic Enzymes"

_processes, doi:10.3390/pr9010038_

Round 1
Reviewer 1 Report
Accept in present form.
Reviewer 2 Report
Attia Iram, Deniz Cekmecelioglu, and Ali Demirci summarized the knowledge about hydrolytic enzymes production with special emphasis on lignocellulolytic enzymes (cellulases, hemicellulases and lignases). A big part of the review article is devoted to lignocellulosic biomass, an ideal feedstock for fungal and bacterial enzymes production. Although the manuscript is written in an interesting way, in my humble opinion the article needs some improvements prior to publication in Processes journal. Below I present my comments and suggestions on “Ideal feedstock and fermentation process improvements for the production of lignocellulolytic enzymes”.
Specific comments:
- In my opinion, it would be an interesting idea to add an additional figure showing the mechanisms of action of enzymes mentioned in the manuscript, i.e. cellulases, hemicellulases and lignases.
- Other constituents of lignocellulosic biomass were omitted in the manuscript. I realize that they are present in small amount in the biomass but may have an impact on the growth of microorganisms. Pectin or pectin-like compounds, as well as proteins, starch or even amino acids, alkaloids and inorganic elements, are present in the lignocellulosic biomass. Please see: Appl.Sci. 2020, 10, 7698 (doi: 10.3390/app10217698).
- Section 2 – please improved this chapter, provide more information about microbial species capable of producing mentioned types of hydrolytic enzymes. In my humble opinion the sentence: “These enzymes are secreted by different white-rot fungal species” is unclear.
- Section 3. “Applications of lignocellulolytic enzymes” should be improved. Please specify the use of these enzymes, show that they are valuable for different industries, include more pros and cons of their use in food and other industries. Moreover, please use more numerical values showing their impact on different properties of food or other products in comparison with untreated enzymatically products.
- Section 6. “Pretreatment of feedstock for lignocellulolytic enzyme production” – Please use the same font and the same size in the whole chapter. More information from Table 3 should be included in the text. Furthermore, other pretreatment types should be found in the literature. What about sonication, organic solvent treatment or use of the combined methods or sequential acid-alkaline pretreatment?
Tables:
- All of the abbreviations in the tables should be defined in the table footers, e.g. DDGS, FPase, CMCase, etc.
- Use the italics when it is necessary, e.g. Populus tremuloides.
- Please add in the tables 3, 4, and 5 names of microorganisms, when you added enzyme activity.
Minor comments:
- Abstract: The aim of the work should be in the past tense.
- "All Figures, Schemes and Tables should be inserted into the main text close to their first citation" - please correct the manuscript according to Instructions for Authors.
- Please improve the references list in journal style.
- Improve also parts of the manuscript referring to genetic modification of microorganisms.
Round 2
Reviewer 2 Report
I would like to thank the authors for their great job in editing the manuscript. The manuscript had been revised carefully and should be accepted after minor revision.
I have only one comment, that the Processes template should be used to prepare your manuscript to publication in Processes journal.